# ADAPTIVE REGULARIZED CUBICS USING QUASI-NEWTON APPROXIMATIONS

## ABSTRACT

Stochastic gradient descent and other first-order variants, such as Adam and Ada-Grad, are commonly used in the field of deep learning due to their computational efficiency and low-storage memory requirements. However, these methods do not exploit curvature information. Consequently, iterates can converge to saddle points and poor local minima. To avoid this, directions of negative curvature can be utilized, which requires computing the second-derivative matrix. In Deep Neural Networks (DNNs), the number of variables ($n$) can be of the order of tens of millions, making the Hessian impractical to store ($\mathcal{O}(n^2)$) and to invert ($\mathcal{O}(n^3)$). Alternatively, quasi-Newton methods compute Hessian approximations that do not have the same computational requirements. Quasi-Newton methods re-use previously computed iterates and gradients to compute a low-rank structured update. The most widely used quasi-Newton update is the L-BFGS, which guarantees a positive semi-definite Hessian approximation, making it suitable in a line search setting. However, the loss function in DNNs are non-convex, where the Hessian is potentially non-positive definite. In this paper, we propose using a Limited-Memory Symmetric Rank-1 quasi-Newton approach which allows for indefinite Hessian approximations, enabling directions of negative curvature to be exploited. Furthermore, we use a modified Adaptive Regularized Cubics approach, which generates a sequence of cubic subproblems that have closed-form solutions. We investigate the performance of our proposed method on autoencoders and feed-forward neural network models and compare our approach to state-of-the-art first-order adaptive stochastic methods as well as L-BFGS.

## 1 INTRODUCTION

Most deep learning problems involve minimization of the empirical risk of estimation

$$\min_{\Theta} f(x; \Theta), \tag{1}$$

where $\Theta \in \mathbb{R}^n$ is the set of weights and $f$ is some scalar-valued loss function. To solve (1), various optimization approaches have been implemented, which we describe below. Throughout this paper, we write $f(\Theta)$ and $f(x; \Theta)$ interchangeably.

**Gradient and adaptive gradient** methods are widely used for training deep neural networks (DNN) for their computational efficiency. The most common approach is Stochastic Gradient Descent (SGD) which, despite its simplicity, performs well over a wide range of applications. However, in a sparse training data setting, SGD performs poorly due to limited training speed (Luo et al. (2019)). To address this problem, *adaptive* methods such as AdaGrad (Duchi et al. (2011)), AdaDelta (Zeiler (2012)), RMSProp (Hinton et al. (2012)) and Adam (Kingma & Ba (2014)) have been proposed. These methods take the root mean square of the past gradients to influence the current step. Amongst all of these adaptive methods, Adam is arguably the most widely used in a deep learning setting due to it rapid training speed.

**Newton's method** has the potential to exploit curvature information from the second-order derivative (Hessian) matrix (see e.g., Gould et al. (2000)). Generally, the iterates are defined by $\Theta_{k+1} = \Theta_k - \alpha_k \nabla^2 f(\Theta_k)^{-1} \nabla f(\Theta_k)$, where $\alpha_k > 0$ is a steplength defined by a linesearch criterion (Nocedal & Wright (2006)). In a DNN setting, we know that the number of parameters ($n$) of the network can be of the order of millions. Thus storing the Hessian which takes $\mathcal{O}(n^2)$

memory, becomes impractical. In addition, the inversion of the Hessian matrix, which takes $\mathcal{O}(n^3)$ operations, is also impractical. Even though Newton's method achieves convergence in fewer steps, the method becomes computationally intractable to use on large-scale DNNs.

**Quasi-Newton** methods are alternatives to Newton methods. They compute Hessian approximations, $\mathbf{B}_{k+1}$, that satisfy the *secant condition* given by $\mathbf{y}_k = \mathbf{B}_{k+1}\mathbf{s}_k$, where $\mathbf{s}_k = \Theta_{k+1} - \Theta_k$ and $\mathbf{y}_k = \nabla f(\Theta_{k+1}) - \nabla f(\Theta_k)$. The most commonly used quasi-Newton method, including in the realm of deep learning, is the limited-memory BFGS update, or L-BFGS (see e.g., Liu & Nocedal (1989)), where the Hessian approximation is given by

$$\mathbf{B}_{k+1} = \mathbf{B}_k + \frac{\mathbf{y}_k\mathbf{y}_k^\top}{\mathbf{y}_k^\top\mathbf{s}_k} - \frac{\mathbf{B}_k\mathbf{s}_k\mathbf{s}_k^\top\mathbf{B}_k^\top}{\mathbf{s}_k^\top\mathbf{B}_k\mathbf{s}_k}. \tag{2}$$

The generic L-BFGS quasi-Newton update scheme is described in Algorithm 1, and numerous variants of L-BFGS exist (see Goldfarb et al. (2020); Moritz et al. (2016); Gower et al. (2016)). One advantage of using an L-BFGS update is that the Hessian approximation can be guaranteed to be definite, which is highly suitable in line-search settings because the update $\mathbf{s}_k$ is guaranteed to be a descent direction, meaning there is some step length along this direction that results in a decrease in the objective function (see Nocedal & Wright (2006), Algorithm 6.1). However, because the L-BFGS update is positive definite, it does not readily detect directions of negative curvature for avoiding saddle points. In contrast, the Symmetric-Rank One (SR1) quasi-Newton update is not guarateed to be positive definite and can result in *ascent* directions for line-search methods. However, in trust-region settings where indefinite Hessian approximations are an advantage because they capture directions of negative curvature, the limited-memory SR1 (L-SR1) has been shown to outperform L-BFGS in DNNs for classification (see Erway et al. (2020)). We discuss this in more detail in Section 2 but in the context of Adaptive Regularization using Cubics.

---

**Algorithm 1** L-BFGS Quasi-Newton Method with Line Search

---

**Require:** Initial weights $\Theta_0$, batch size $d$, learning rate $\alpha$, dataset $\mathcal{D}$, loss function $f(\Theta)$.
    **for** $k = 0, 1, 2, \ldots$ **do**
        Sample mini-batch of size $d : \mathcal{D}_k \subseteq \mathcal{D}$
        Perform the forward backward pass over the current mini-batch
        Compute the limited memory approximation $B_k$ using (2)
        Compute step $\mathbf{s}_k = \alpha\mathbf{B}_k^{-1}\nabla_\Theta f(\Theta_k)$, where $\alpha$ is the line-search step length
    **end for**

---

## 2   L-SR1 Adaptive Regularization using Cubics Method

We begin by discussing the L-SR1 update and the adaptive regularizion using cubics methods for large-scale optimization.

Unlike the BFGS update (2), which is a rank-two update, the **SR1 update** is a rank-one update, which is given by

$$\mathbf{B}_{k+1} = \mathbf{B}_k + \frac{1}{\mathbf{s}_k^\top(\mathbf{y}_k - \mathbf{B}_k\mathbf{s}_k)}(\mathbf{y}_k - \mathbf{B}_k\mathbf{s}_k)(\mathbf{y}_k - \mathbf{B}_k\mathbf{s}_k)^\top \tag{3}$$

(see Khalfan et al. (1993)). As previously mentioned, $\mathbf{B}_{k+1}$ in (3) is not guaranteed to be definite. However, it can be shown that the SR1 matrices can converge to the true Hessian (see Conn et al. (1991) for details). We note that the pair $(\mathbf{s}_k, \mathbf{y}_k)$ is accepted only when $|\mathbf{s}_k^\top(\mathbf{y}_k - \mathbf{B}_k\mathbf{s}_k)| > \varepsilon\|\mathbf{y}_k - \mathbf{B}_k\mathbf{s}_k\|_2^2$, for some constant $\varepsilon > 0$ (see Nocedal & Wright (2006), Sec. 6.2, for details). The SR1 update can be defined recursively as

$$\mathbf{B}_{k+1} = \mathbf{B}_0 + \sum_{j=0}^{k} \frac{1}{\mathbf{s}_j^\top(\mathbf{y}_j - \mathbf{B}_j\mathbf{s}_j)}(\mathbf{y}_j - \mathbf{B}_j\mathbf{s}_j)(\mathbf{y}_j - \mathbf{B}_j\mathbf{s}_j)^\top. \tag{4}$$

In limited-memory SR1 (L-SR1) settings, only the last $m \ll n$ pairs of $(\mathbf{s}_j, \mathbf{y}_j)$ are stored and used. If $\mathbf{S}_{k+1} = [\ \mathbf{s}_0 \ \ \mathbf{s}_1 \ \ \cdots \ \ \mathbf{s}_k\ ]$ and $\mathbf{Y}_{k+1} = [\ \mathbf{y}_0 \ \ \mathbf{y}_1 \ \ \cdots \ \ \mathbf{y}_k\ ]$, then $\mathbf{B}_{k+1}$ admits a **compact**

**representation** of the form

$$\mathbf{B}_{k+1} \; = \; \mathbf{B}_0 + \begin{bmatrix} \boldsymbol{\Psi}_{k+1} \end{bmatrix} \begin{bmatrix} \mathbf{M}_{k+1} \end{bmatrix} \begin{bmatrix} \boldsymbol{\Psi}_{k+1}^\top \end{bmatrix}, \tag{5}$$

where

$$\boldsymbol{\Psi}_{k+1} = \mathbf{Y}_{k+1} - \mathbf{B}_0 \mathbf{S}_{k+1} \quad \text{and} \quad \mathbf{M}_{k+1} = (\mathbf{D}_{k+1} + \mathbf{L}_{k+1} + \mathbf{L}_{k+1}^\top - \mathbf{S}_{k+1}^\top \mathbf{B}_0 \mathbf{S}_{k+1})^{-1}, \tag{6}$$

where $\mathbf{L}_{k+1}$ is the strictly lower triangular part, $\mathbf{U}_{k+1}$ is the strictly upper triangular part, and $\mathbf{D}_{k+1}$ is the diagonal part of $\mathbf{S}_{k+1}^\top \mathbf{Y}_{k+1} = \mathbf{L}_{k+1} + \mathbf{D}_{k+1} + \mathbf{U}_{k+1}$ (see Byrd et al. (1994) for further details).

Because of the compact representation of $\mathbf{B}_{k+1}$, its partial eigendecomposition can be computed (see Burdakov et al. (2017)). In particular, if we compute the QR decomposition of $\boldsymbol{\Psi}_{k+1}$, then we can write $\mathbf{B}_{k+1} = \mathbf{B}_0 = \mathbf{U}_\| \hat{\boldsymbol{\Lambda}}_{k+1} \mathbf{U}_\|^\top$, where $\mathbf{U}_\| \in \mathbb{R}^{n \times (k+1)}$ has orthonormal columns and $\hat{\boldsymbol{\Lambda}} \in \mathbb{R}^{(k+1) \times (k+1)}$ is a diagonal matrix. If $\mathbf{B}_0 = \delta_k \mathbf{I}$ (see e.g., Lemma 2.4 in Erway et al. (2020)), where $0 < \delta_k < \delta_{\max}$ is some scalar and $\mathbf{I}$ is the identity matrix, then we obtain the eigendecomposition $\mathbf{B}_{k+1} = \mathbf{U}_{k+1} \boldsymbol{\Lambda}_{k+1} \mathbf{U}_{k+1}^\top$, where $\mathbf{U}_{k+1} = [\mathbf{U}_\| \;\; \mathbf{U}_\perp]$, with $\mathbf{U}_\perp \in \mathbb{R}^{n \times (n - (k+1))}$ and $\mathbf{U}_{k+1}^\top \mathbf{U}_{k+1} = \mathbf{I}$. Here, $(\boldsymbol{\Lambda}_{k+1})_i = \delta_k + \hat{\lambda}_i$ for $i \leq k + 1$, where $\hat{\lambda}_i$ is the $i$th diagonal in $\hat{\boldsymbol{\Lambda}}_{k+1}$, and $(\boldsymbol{\Lambda})_i = \delta_k$ for $i > k + 1$.

Since the SR1 Hessian approximation can be indefinite, some safeguard must be implemented to ensure that the resulting search direction $\mathbf{s}_k$ is a descent direction. One such safeguard is to use a "regularization" term.

The **Adaptive Regularization using Cubics (ARCs)** method (Griewank (1981); Cartis et al. (2011)) can be viewed as an alternative to line-search and trust-region methods. At each iteration, an approximate global minimizer of a local (cubic) model,

$$\min_{\mathbf{s} \in \mathbb{R}^n} m_k(\mathbf{s}) \equiv \mathbf{g}_k^\top \mathbf{s} + \frac{1}{2} \mathbf{s}^\top \mathbf{B}_k \mathbf{s} + \frac{\mu_k}{3} (\Phi_k(\mathbf{s}))^3, \tag{7}$$

is determined, where $\mathbf{g}_k = \nabla f(\Theta_k)$, $\mu_k > 0$ is a regularization parameter, and $\Phi_k$ is a function (norm) that regularizes $\mathbf{s}$. Typically, the Euclidean norm is used. In this work, we propose an alternative "shape-changing" norm that allows us to solve each subproblem (7) exactly. This shape-changing norm was proposed in Burdakov et al. (2017), and it is based on the partial eigendecomposition of $\mathbf{B}_k$. Specifically, if $\mathbf{B}_k = \mathbf{U}_k \boldsymbol{\Lambda}_k \mathbf{U}_k^\top$ is the eigendecomposition of $\mathbf{B}_k$, then we can define the norm $\|\mathbf{s}\|_{\mathbf{U}_k} \overset{\text{def}}{=} \|\mathbf{U}_k^\top \mathbf{s}\|_3$. Applying a change of basis with $\bar{\mathbf{s}} = \mathbf{U}_k^\top \mathbf{s}$ and $\bar{\mathbf{g}}_k = \mathbf{U}_k^\top \mathbf{g}_k$, we can redefine the cubic subproblem as

$$\min_{\bar{\mathbf{s}} \in \mathbb{R}^n} \bar{m}_k(\mathbf{s}) = \bar{\mathbf{g}}_k^\top \bar{\mathbf{s}} + \frac{1}{2} \bar{\mathbf{s}}^\top \boldsymbol{\Lambda}_k \bar{\mathbf{s}} + \frac{\mu_k}{3} \|\bar{\mathbf{s}}\|_3^3. \tag{8}$$

With this change of basis, we can find a closed-form solution of (8) easily. The proposed Adaptive Regularization using Cubics with L-SR1 (ARCSLSR1) algorithm is given in Algorithm 2.

## 2.1 CONTRIBUTIONS

The main contributions of this paper are as follows: 1. L-SR1 quasi-Newton methods: The most commonly used quasi-Newton approach is the L-BFGS method. In this work, we use the L-SR1 update to better model potentially indefinite Hessians of the non-convex loss function. 2. Adaptive Regularization using Cubics (ARCs): Given that the quasi-Newton approximation is allowed to be indefinite, we use an Adaptive Regularized using Cubics approach to safeguard each search direction. 3. Shape-changing regularizer: We use a shape-changing norm to define the cubic regularization term, which allows us to compute the closed form solution to the cubic subproblem (7). 4. Computational complexity: Let $m$ be the number of previous iterates and gradients stored in memory. The proposed LSR1 ARC approach is comparable to L-BFGS in terms of storage and compute complexity (see Table 1).

---

**Algorithm 2** Limited-Memory Symmetric Rank-1 Adaptive Regularization using Cubics

---
1: **Given:** $\Theta_0, \gamma_2 \geq \gamma_1, 1 > \eta_2 \geq \eta_1 > 0,$ and $\sigma_0 > 0$
2: **for** $k = 0, 1, 2 \ldots$ **do**
3:     Obtain $\mathbf{S}_k = [\, \mathbf{s}_0 \ \cdots \ \mathbf{s}_k \,], \mathbf{Y}_k = [\, \mathbf{y}_0 \ \cdots \ \mathbf{y}_k \,]$
4:     Solve the generalized eigenproblem $\mathbf{S}_k^\top \mathbf{Y}_k \mathbf{u} = \hat{\Lambda} \mathbf{S}_k^\top \mathbf{S}_k \mathbf{u}$ and let $\delta_k = \min\{\hat{\lambda}_i\}$
5:     Compute $\boldsymbol{\Psi}_k = \mathbf{Y}_k - \delta_k \mathbf{S}_k$
6:     Perform QR decomposition of $\boldsymbol{\Psi} = \mathbf{Q}\mathbf{R}$
7:     Compute the eigendecomposition $\mathbf{R}\mathbf{M}\mathbf{R}^\top = \mathbf{P}\boldsymbol{\Lambda}\mathbf{P}^\top$
8:     Assign $\mathbf{U}_\| = \mathbf{Q}\mathbf{P}$ and $\mathbf{U}_\|^\top = \mathbf{P}^\top \mathbf{Q}^\top$
9:     Define $\mathbf{C}_\| = \mathrm{diag}(c_1, \ldots, c_m)$, where $c_i = \frac{2}{\lambda_i + \sqrt{\lambda_i^2 + 4\mu|\bar{\mathbf{g}}_i|}}$ and $\bar{\mathbf{g}}_\| = \mathbf{U}_\|^\top \mathbf{g}$
10:    Compute $\alpha^* = \frac{2}{\delta_k + \sqrt{\delta_k^2 + 4\mu\|\mathbf{g}_\perp\|}}$ where $\mathbf{g}_\perp = \mathbf{g} - \mathbf{U}_\|\bar{\mathbf{g}}_\|$
11:    Compute step $\mathbf{s}^* = -\alpha^* \mathbf{g} + \mathbf{U}_\|(\alpha^* \mathbf{I}_m - \mathbf{C}_\|)\mathbf{U}_\|^\top$
12:    Compute $m(\mathbf{s}^*)$ and $\rho_k = (f(\Theta_k) - f(\Theta_{k+1}))/m(\mathbf{s}^*)$
13:    Set

$$\Theta_{k+1} = \begin{cases} \Theta_k + \mathbf{s}_k, & \text{if } \rho_k \geq \eta_1, \\ \Theta_k, & \text{otherwise} \end{cases} \quad \text{and} \quad \mu_{k+1} = \begin{cases} 0.5\mu_k & \text{if } \rho_k > \eta_2, \\ 0.5\mu_k(1 + \gamma_1) & \text{if } \eta_1 \leq \rho_k \leq \eta_2, \\ 0.5\mu_k(\gamma_1 + \gamma_2) & \text{otherwise} \end{cases}$$

14: **end for**

---

Table 1: Storage and compute complexity of the methods used in our experiments.

| Algorithms | Storage complexity | Compute complexity |
|---|---|---|
| SGD/Adaptive methods | $\mathcal{O}(n)$ | $\mathcal{O}(n)$ |
| L-BFGS | $\mathcal{O}(n + mn)$ | $\mathcal{O}(mn)$ |
| ARCs-LSR1 | $\mathcal{O}(n + mn)$ | $\mathcal{O}(m^3 + 2mn)$ |

## 2.2 IMPLEMENTATION

Because full gradient computation is very expensive to perform, we impement a stochastic version of the proposed ARCs-LSR1 method. In particular, we use the batch gradient approximation

$$\tilde{\mathbf{g}}_k \equiv \frac{1}{|\mathcal{B}_k|} \sum_{i \in \mathcal{B}_k} \nabla f_i(\Theta_k).$$

In defining the SR1 matrix, we use the quasi-Newton pairs $(\mathbf{s}_k, \tilde{\mathbf{y}}_k)$, where $\tilde{\mathbf{y}}_k = \tilde{\mathbf{g}}_{k+1} - \tilde{\mathbf{g}}_k$ (see e.g., Erway et al. (2020)).

## 3 CONVERGENCE ANALYSIS

In this section, we prove convergence properties of the proposed method (ARCs-LSR1 in Algorithm 2). The following theoretical guarantees follow the ideas from Cartis et al. (2011) and Benson & Shanno (2018).

First, we make the following mild assumptions:

**A1.** The loss function $f(\Theta)$ is continuously differentiable, i.e., $f \in C^1(\mathbb{R}^n)$.

**A2.** The loss function $f(\Theta)$ is bounded below.

Next, we prove that the matrix $\mathbf{B}_k$ in (4) is bounded.

**Lemma 1** *The SR1 matrix* $\mathbf{B}_{k+1}$ *in (4) satsifies*

$$\|\mathbf{B}_{k+1}\|_F \leq \kappa_B \ \text{ for all } k \geq 1$$

*for some* $\kappa_B > 0$.

*Proof:* Using the limited-memory SR1 update with memory parameter $m$ in (4), we have

$$\|\mathbf{B}_{k+1}\|_F \leq \|\mathbf{B}_0\|_F + \sum_{j=k-m+1}^{k} \frac{\|(\mathbf{y}_j - \mathbf{B}_j\mathbf{s}_j)(\mathbf{y}_j - \mathbf{B}_j\mathbf{s}_j)^\top\|_F}{|\mathbf{s}_j^\top(\mathbf{y}_j - \mathbf{B}_j\mathbf{s}_j)|}.$$

Using a property of the Frobenius norm, namely, for real matrices $\mathbf{A}$, $\|\mathbf{A}\|_F^2 = \text{trace}(\mathbf{A}\mathbf{A}^\top)$, we have that $\|(\mathbf{y}_j - \mathbf{B}_j\mathbf{s}_j)(\mathbf{y}_j - \mathbf{B}_j\mathbf{s}_j)^\top\|_F = \|\mathbf{y}_j - \mathbf{B}_j\mathbf{s}_j\|_2^2$. Since the pair $(\mathbf{s}_j, \mathbf{y}_j)$ is accepted only when $|\mathbf{s}_j^\top(\mathbf{y}_j - \mathbf{B}_j\mathbf{s}_j)| > \varepsilon\|\mathbf{y}_j - \mathbf{B}_j\mathbf{s}_j\|_2^2$, for some constant $\varepsilon > 0$, and $\mathbf{B}_0 = \delta_k\mathbf{I}$ for some $0 < \delta_k < \delta_{\max}$, we have

$$\|\mathbf{B}_{k+1}\|_F \leq \delta_{\max} + \frac{m}{\varepsilon} \equiv \kappa_B.$$

$\square$

Given the bound on $\|\mathbf{B}_{k+1}\|_F$, we obtain the following result, which is similar to Theorem 2.5 in Cartis et al. (2011).

**Theorem 1** *Under Assumptions A1 and A2, if Lemma 1 holds, then*

$$\lim_{k\to\infty}\inf \|\mathbf{g}_k\| = 0.$$

Finally, we consider the following assumption, which can be satisfied when the gradient, $\mathbf{g}(\Theta)$, is Lipschitz continuous on $\Theta$.

**A3.** If $\{\Theta_{t_i}\}$ and $\{\Theta_{l_i}\}$ are subsequences of $\{\Theta_k\}$, then $\|\mathbf{g}_{t_i} - \mathbf{g}_{l_i}\| \to 0$ whenever $\|\Theta_{t_i} - \Theta_{l_i}\| \to 0$ as $i \to \infty$. If we further make Assumption **A3**, we have the following stronger result (which is based on Corollary 2.6 in Cartis et al. (2011)):

**Corollary 1** *Under Assumptions A1, A2, and A3, if Lemma 1 holds, then*

$$\lim_{k\to\infty} \|\mathbf{g}_k\| = 0.$$

## 4 EXPERIMENTS

To empirically compare the efficiency of the method against popular optimization methods like SGD, ADAGRAD, ADAM, RMSProp and L-BFGS, we focus on two broad deep learning problems: image classification and image reconstruction. We choose these tasks due to their broad importance and availability of reproducible model architectures. We run each experiments on an average of 5 times with a random initialization in each experiment. The number of parameters, convolutional layers and fully connected layers are mentioned in Table 3.

**Dataset:** We measure the classification performance of each optimization method on 4 image datasets: MNIST (LeCun et al. (2010)), FashionMNIST (Xiao et al. (2017)), IRIS (Dua & Graff (2017)) and CIFAR10 (Krizhevsky et al.). We have provided a comprehensive view of the experiments in Table 2

| Dataset | Network | Type |
|---------|---------|------|
| IRIS | MLP | Classification |
| MNIST | MLP | Classification |
| FMNIST | Convolutional | Classification |
| CIFAR10 | Convolutional | Classification |
| FashionMNIST | Convolutional | Reconstruction |
| MNIST | Convolutional | Reconstruction |

Table 2: List of experiments

**Hyperparameter tuning:** We empirically fine-tune the hyperparameters and select the best for each update scheme. We have made a comprehensive list of all the learning rates for the gradient and adaptive gradient based algorithms in Table 4 in the Appendix. The additional parameters are defined as follows:

- ADAM: We apply an $\epsilon$ perturbation of $1.0 \times 10^{-6}$. $\beta_0$ and $\beta_1$ are chosen to be 0.9 and 0.999, respectively.
- ADAGRAD: The initial accumulator value is set to 0. The perturbation $\epsilon$ is set to $1.0 \times 10^{-10}$.
- SGD: We use a momentum of 0.9.
- RMSPROP: We set $\alpha = 0.99$. The perturbation $\epsilon$ is set $1.0 \times 10^{-8}$.
- L-BFGS: The table 4 in Appendix A presents the *initial* learning rate for the stochastic step in L-BFGS. We set the default learning rate to 1.0. We choose a history size $m$ of 10 and max iterations to 10. The tolerance on function value/parameter change is set to $1.0 \times 10^{-9}$ and the first-order optimality condition for termination is defined as $1.0 \times 10^{-9}$.
- ARC-LSR1: We choose the same parameters as L-BFGS.

**Network architecture:** For each problem, we define the model architecture in Table 3 in the appendix. We define the process of the forward and backward pass of a DNN in Algorithm 3 in the appendix.

| Dataset | Network | Convolution layers | Fully connected layers | Parameters |
|---|---|---|---|---|
| IRIS | Classifier | - | 3 | 2953 |
| MNIST | Classifier | - | 3 | 397510 |
| CIFAR10 | Classifier | 2 | 3 | 62006 |
| MNIST | Autoencoder | 6 | 4 | 53415 |
| FashionMNIST | Autoencoder | 6 | 4 | 53415 |

Table 3: List of experiments

**Testbed and software:** All experiments were conducted using open-source software PyTorch (Paszke et al. (2019)), SciPy (Virtanen et al. (2020)) and NumPy (Harris et al. (2020)). We use an Intel Core i7-8700 CPU with a clock rate of 3.20 GHz and an NVIDIA RTX 2080 Ti graphics card.

## 5 RESULTS

We have divided the sections into two categories: classification and image reconstruction. We present both the training results and the testing results for all methods.

### 5.1 CLASSIFICATION RESULTS

For each classification problem, we define the network architecture, the corresponding hyperparameters (other than the learning rate) for each optimization scheme.

**IRIS:** Since this dataset is relatively small, we assume a small network for our deep-learning model. The model is described in 3. We set the history size for the proposed approach and L-BFGS to 10 and the number of iterations to 10. Figure 1 shows the comparative performance of all the methods. Note that our proposed method (ARCLSR1) achieves the highest classification accuracy in the fewest number of epochs.

**MNIST:** We trained the network for 20 epochs with a batch size of 256 images each. We keep the same history size and number of iterations as the IRIS dataset for L-BFGS and the proposed ARCLSR1 approach. For training, it can be seen in Figure 2 that nearly all methods achieve optimal training accuracy. However, closely inspecting the testing curve, we notice that the proposed approach achieves higher accuracy than all the existing methods.

**FMNIST:** We train the network for 20 epochs with a batch size of 256 images. We keep the history size the same as the IRIS and MNIST experiments for the proposed approach and L-BFGS. For this method, the proposed ARCLSR1 approach is comparable to L-BFGS but outperforms the adaptive methods (see Figure 3).

**CIFAR10:** We use the same parameters presented in Table 4 in the previous section for the adaptive methods. For ARCLSR1 and L-BFGS, we have a history size of 100 with a maximum number of iterations of 100 and a batch size of 1024. Figure 4(a) represents the training loss (cross-entropy

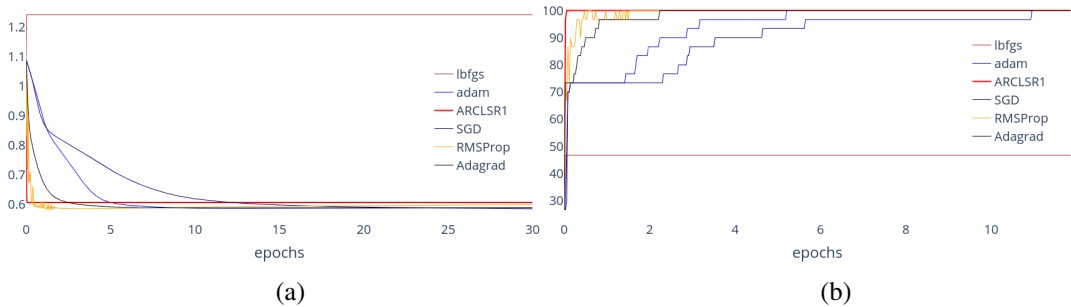

(a)                 (b)

Figure 1: The classification results on the IRIS dataset. (a) Training loss of the network. The $y$-axis represents the negative log-likelihood loss and the $x$-axis represents the number of epochs. (b) The classification accuracy for each method, i.e., the percentage of testing samples correctly predicted in the testing dataset.

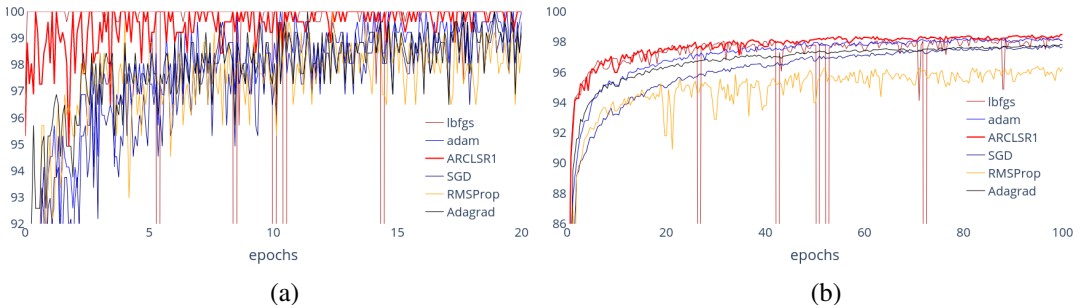

(a)                 (b)

Figure 2: The classification results on MNIST. The $y$-axis represents the classification accuracy on the MNIST dataset, and the $x$-axis represents the number of epochs. (a) Training response. (b) Testing response.

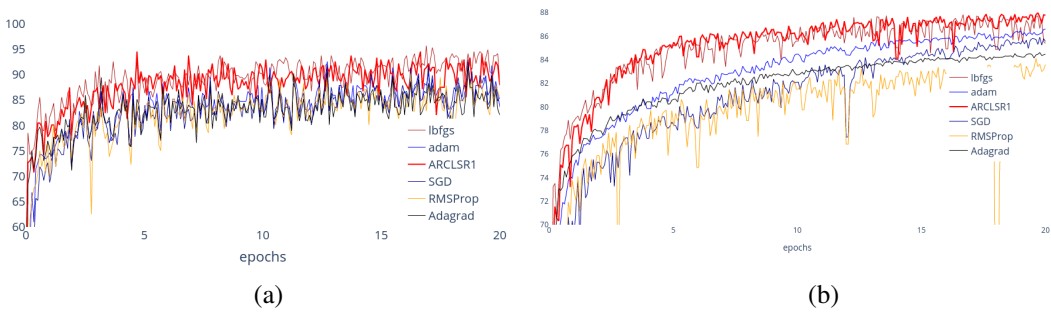

(a)                 (b)

Figure 3: The plots above show the classification results on the Fashion MNIST dataset. We run this experiment for 20 epochs. In this experiment, the proposed method is comparable to L-BFGS but outperforms the adaptive methods.

loss). Figure 4(b) represents the testing accuracy, i.e., number of sample correctly predicted in the testing set. To demonstrate the efficacy of the proposed method on larger networks, additional experimentation on the ResNet50 architecture can be found in the appendix (Figure 8).

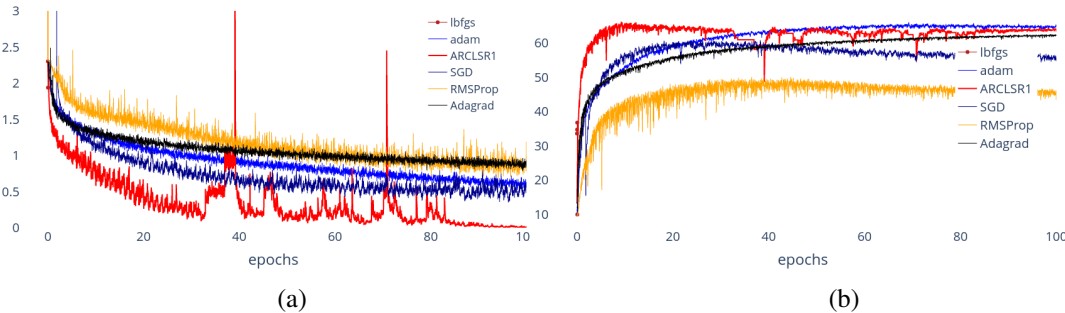

(a)                                             (b)

Figure 4: The classification results on CIFAR10. (a) The $y$-axis represents the training cross-entropy loss, and the $x$-axis represents the number of batches. (b) The $y$-axis represents the testing response accuracy and the $x$-axis represents the number of epochs.

## 5.2 IMAGE RECONSTRUCTION RESULTS

The image reconstruction problem involves feeding a feedforward convolutional autoencoder model (with randomly initialized weights) a batch of the dataset. It follows the same deep learning convention as mentioned in Algorithm 3 in Appendix A. The loss function is defined between the reconstructed image and the original image.

**MNIST:** An image $x \in \mathbb{R}^n$ is fed to the network, compressed into a latent space $z \in \mathbb{R}^l$, where $l \ll n$, and reconstructed back to its original image size $\bar{x} \in \mathbb{R}^n$. We compute the mean-squared loss error between the reconstruction and the true image. The weights are initialized randomly. Each experiment has been conducted 5 times and we considered a batch size of 256 images each with 50 epochs. The results for the image reconstruction can be seen in Figure 5.

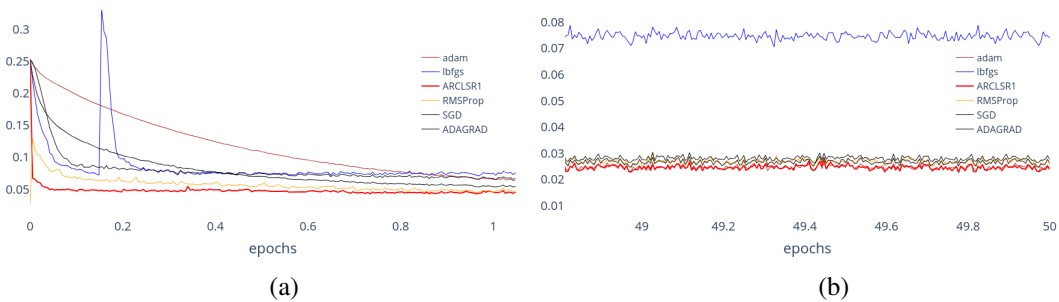

(a)                                             (b)

Figure 5: This graph represents the training accuracy on the training samples. The $x$-axis shows the number of epochs and the $y$-axis represents the accuracy (Mean-Squared error). (a) shows the initial training error and (b) shows the final training error.

One can notice that the initial descent provided by the proposed approach provides a significant decrease in the objective function. To understand better, we provide the details of the results during the initial epoch (Figure 9(a)) and the final epoch (Figure 9(b)). We notice that the ARCLSR1 method has minimized efficiently in the first half of the first epoch. This is empirical evidence that the method converges to the minimizer in fewer steps in comparison to the adaptive methods. In Figure 5 (b), we notice that all the adaptive methods eventually converge to the same point. For training response results on the F-MNIST dataset, see Section B in the appendix.

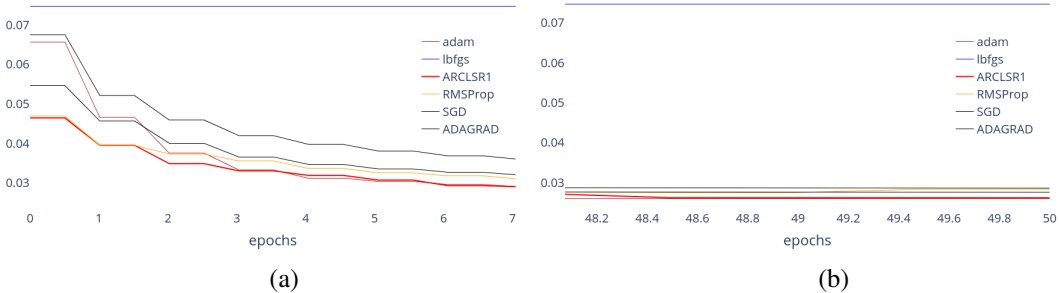

(a)                                                    (b)

Figure 6: The testing accuracy on MNIST dataset. $y$-axis represents the Mean-Squared Error loss on the testing set and $x$-axis represents the number of epochs. (a) shows the initial testing response. (b) shows the final testing response.

## 5.3 TIME COMPLEXITY ANALYSIS

We understand that the proposed approach performs competitively against all existing methods. We now analyze the time-constraints of each method. We choose to clock the computationally demanding algorithm here - CIFAR10 classification. We chose a maximum iterations of 100 with a history size of 100 for L-BFGS and the ARCs LSR1, with a batch size of 1024 images. Figure 7 plots the time required by each of the methods to reach non-overtrained minima with a batch size of 1024 images. As can be seen, the proposed approach is able to reach the desired minima in much less time than the rest of the algorithms. L-BFGS finds it hard to converge due to a very noisy loss function and a small batch size, thus causing the algorithm to break. Ozyildirim & Kiran (2020) argue that a large batch size is required for quasi-Newton methods to perform well. However, the ARCLSR1 method performs well with a small batch size as well.

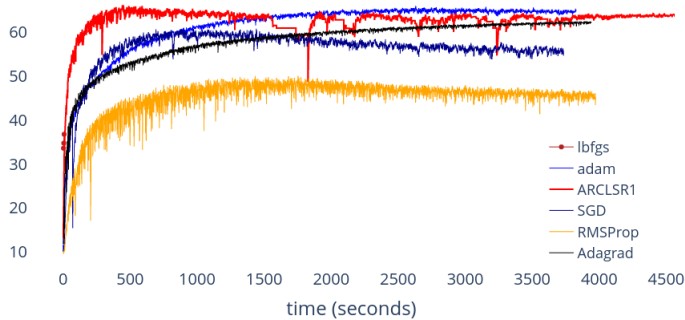

Figure 7: Timing analysis for CIFAR10. (a) Evolution of model accuracy with respect to time ($x$-axis is time in seconds and $y$-axis is accuracy of prediction in percentage). (b) Computational cost for each epoch (the $x$-axis corresponds to epochs and the $y$-axis is time).

## 6 CONCLUSION

In this paper, we proposed a novel quasi-Newton approach in a modified adaptive regularized cubic setting. We were able to empirically and theoretically show how an L-SR1 quasi-Newton approximation in an ARCs setting was able to perform either better or comparably to most of the state of the art optimization schemes. Even though the approach has yielded exceptional results, we need to test the method's efficacy when the network size and dataset size is large and when availability of data is sparse.

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
