# OpenReview forum: "L-SR1 Adaptive Regularization by Cubics for Deep Learning"
_ICLR.cc/2022/Conference — ICLR 2022 Submitted_

### Official Review · Reviewer_cfGs · 2021-10-27

**Correctness:** 2
**Technical Novelty And Significance:** 2
**Empirical Novelty And Significance:** 1
**Recommendation:** 3
**Confidence:** 4

**Main Review:**

Pros:
- The paper highlights the potential of L-SR1 in the highly non-convex deep neural network setting, where neglecting negative curvature information might seriously degrade the accuracy of the Hessian approximation.
- The idea of using the specific type of shape-changing norm that varies in every iteration in order to simplify the local cubic sub-problem is interesting.

Cons:
- The novelty of this paper is very limited. The only novel contribution compared to Cartis et al.* and Erway et al*. seems to be the particular form of shape-changing regularization. This has the potential to be a valuable contribution, but the exposition of the technique is very confusing. Authors point out that that closed-form solution can be found easily, but provide no further explanation, simply presenting the full algorithm as is. The proposed algorithm, which is the main contribution of this paper, is not discussed in much detail and several steps are left unexplained.
- The significance of the results is questionable. Since the paper does not provide any theoretical insights into the introduced algorithm (e.g. convergence guarantees), it has to be thoroughly verified through relevant experiments. The proposed algorithm is not compared to its closest competitor, Erway et al, and the L-BFGS algorithm used in the comparison is a naive implementation of mini-batch L-BFGS that is known to be unstable on larger models/datasets. There are several improved versions of L-BFGS tailored to the stochastic setting (Moritz et al.*, Gower et al.* and more). Furthermore, the experiments are performed on very small and simple models and datasets, casting doubt on the practical relevance of the method. Even in this limited setting the gains over other techniques are not too significant.
- Authors posit that Newton's method avoid saddle points by exploiting curvature information. To the best of my knowledge, the opposite is true and Newton's method is prone to converging to saddle points. The motivation that SR1 avoids saddle points by using negative curvature information is similarly unsupported. It seems to me that the benefit of SR1 over BFGS would lie in the more accurate Hessian representation and not in the avoidance of saddles.
- The structure and organization of the paper is confusing and has unnecessary parts. For instance I don't see why the specific forms of adaptive methods are discussed in such detail (including table) when it is not used later in the paper. Including forward and backward pass of a DNN as an algorithm is also unnecessary at a conference like ICLR. Detailed description of the well-known datasets, and the very detailed discussion of minor hyperparameters could be moved to the appendix. The paper has many typos, grammatical errors, unfinished sentences.
- The experimental results are confusing and inconsistent. In some plots we don't know exactly what quantity is plotted (Fig. 3). In some plots, epochs are not integers, which is confusing and in this case iterations would be more clear. Figure 7 shows 'computational cost' but it is not disclosed how authors calculated this quantity. In the same plot CIFAR-10 training on a small model takes 80k seconds (approximately 22 hours), which seems highly unrealistic. In 'Conclusions' authors also mention that the proposed algorithm's performance is fairly independent of batch size, which is not supported at all by the experiments. Overall, the experiments have brought me more questions than answers.

*Citations are equivalent to those in the paper.

**Summary Of The Paper:**

This paper investigates the application of a certain Quasi-Newton algorithm, the Limited-Memory Symmetric Rank-1 (L-SR1) algorithm, in deep learning problems. The benefit of this technique over similar more widely investigated methods that use a positive definite approximation of the Hessian, such as stochastic L-BFGS, is the fact that the L-SR1 approximation is not guaranteed to be definite, and thus has the potential to have a more accurate approximation of the true Hessian. Since in this case line-search methods may return ascent directions, authors propose a specific form of Adaptive Regularization using Cubics (ARC) as an alternative. Numerical simulations are provided comparing the performance of the proposed algorithm to SGD, adaptive methods such as Adam and a naive L-BFGS implementation.

**Summary Of The Review:**

Even though the paper has a neat idea to transform the ARC subproblem into an easily solvable form, the paper has many shortcomings when it comes to novelty and significance to the ML community detailed above. At this point the paper reads more as a draft, and would require very significant modifications in my opinion to improve its quality. Therefore I recommend rejection. I am however open to update my score if the authors or other reviewers address my concerns.

---

> ### Author Response · Authors · 2021-11-23
> **Official Rebuttal of Paper4406 by Author: Revisions**
>
> We thank the reviewer for taking the time to read and review the paper. We have added further explanation in the new draft (kindly refer to Section 2: LSR-1 Adaptive Regularization using Cubics, Section 2.1: Contributions and section 3 Convergence analysis).
> For the comparison of results, we have only used methods which are reproducible and readily available.  In particular, we did not compare to Erway et al because the code was not publicly available.
> We should have been more precise in our description of Newton’s method.  Specifically, there are variants of Newton’s method that exploit directions of negative curvature to avoid saddle points (see e.g., “Exploiting negative curvature directions in linesearch methods for unconstrained optimization” by Gould, Lucidi, Roma, and Toint (2008), and the Notes and References in Sec. 3.5 of “Numerical Optimization” by Nocedal and Wright). Regarding the SR1 method and negative curvatures, in Lemma 2.3 of the Erway et al paper, it is stated that if the SR1 matrix is not positive semi-definite, then as the trust-region increases, the trust-region solution becomes more parallel to the eigenspace corresponding to the most negative eigenvalue of the SR1 matrix.  So in the end, the trust-region solution is influenced by this eigenspace if the SR1 matrix has a negative eigenvalue.
> We have revised the structure of the paper and moved all the supplementary information to the appendix. This includes the forward and backward pass, detailed explanation of the hyper parameters, and network architecture.
> We have revised all the graphs to fix the inconsistencies in the revised paper. We have removed the line that the method is ‘independent of batch sizes’ as it is not clear from the paper. However, we have chosen the optimal batch size for all the competing methods.

---

> > ### Comment · Reviewer_cfGs · 2021-11-29
> > **Response to authors**
> >
> > I thank the authors for addressing my concerns and for clarifying their point with respect to saddle points in quasi-Newton methods. However, I still believe that the paper has limited novelty, and that the proposed algorithm should be compared with more competitive, stochastic variants of quasi-Newton optimizers. Furthermore, results on more practical models (e.g. ResNet) and larger datasets would further improve the confidence in the practical potential of the proposed optimizer. Overall, I am keeping my score.

---

### Official Review · Reviewer_oGs7 · 2021-11-02

**Correctness:** 2
**Technical Novelty And Significance:** 2
**Empirical Novelty And Significance:** 3
**Recommendation:** 3
**Confidence:** 4

**Main Review:**

My major concerns with this paper are :
-Except thm1 from Reddi et al. paper, I dont see any theoretical result neither in the paper nor in the  appendix. Am I missing some thing or no theory is given to support the claims in the paper?
For instance, after Thm1 statement, you claim that Newton's method avoid saddle points...but I did not see any theoretical results supporting this...
-The novelty of the contributions stated at page 4: the methods SR1 to approximate the Hessian & ARC  exist already in the literature.

You claim at the end of page 2 "... an L-BFGS update is that the Hessian approximation can be guaranteed to be definite, which is highly suitable in line-search settings because the update s_k is guaranteed to be a descent direction..." why? give a reference or show it?

Line 4 in Algo 1: what is the operator G?

what is f_t?

Thm1: what do you mean by online convex optimization exactly?

Several typos in the paper.
...

**Summary Of The Paper:**

The paper proposes to use a Symmetric Rank-1 (SR1) quasi-Newton approach to approximate the Hessian and  to use an Adaptive Regularized Cubics (ARC) with an adaptive norm framework. To assess the performance of the new method, numerical experiments using autoencoders and feed-forward neural network models are supplied.

**Summary Of The Review:**

See above

---

> ### Author Response · Authors · 2021-11-23
> **Official Rebuttal of Paper4406 by Author: Revisions**
>
> We thank the reviewer for their time and effort to read our paper. As per the reviewers wishes, we have provided the convergence guarantees of the proposed approach. We have also added the reference that shows that Newton’s methods avoid saddle points.
>
> We have provided the reference to show the suitability of the L-BFGS method to a line search setting.
>
> We have removed Theorem 1 and Algorithm 1 as it did not have much relevance to the paper.
>
> We have reread the papers and have corrected the typos that we identified.

---

### Official Review · Reviewer_YQSL · 2021-11-02

**Correctness:** 3
**Technical Novelty And Significance:** 3
**Empirical Novelty And Significance:** 2
**Recommendation:** 5
**Confidence:** 4

**Main Review:**

The ingredients in the algorithm are not new, though the shape-changing regularizer used in this paper was originally described for a trust-region method rather than ARC (and the author of the shape-changing regularizer paper focused on L-BFGS as his main algorithm, while at the same time mentioning L-SR1 and other quasi-Newton methods as options).  However, the particular combination is new as far as I know, and the empirical results seem quite promising, though all involve relatively small networks.

To the extent that the combination is new, I would have liked to see a little more rationale for the details of the regularizer adjustment and step acceptance criteria.  These seem to be adapted from the paper of Burdakov to the ARC setting, but there is no discussion of convergence results.  If the algorithm is the main contribution of the paper, a statement of these results would be nice to have.  It would also be nice to have the (very short) appendix A folded into the main paper text so that the paper could be implemented by a reader without referring to the supplementary material.

The text is missing some critical details that I think make the algorithm much more interesting (and make me interested in seeing a more developed paper in the future!).

First, it seems that the L-SR1 and L-BFGS algorithms are both being used with stochastic gradient estimates in lieu of actual gradients.  This is not entirely clear in the earlier examples where the authors refer to mini-batch sizes (those could have just been for the SGD variants), but in the final paragraph of Section 4, just before the conclusion, the authors refer to comparing L-SR1 and L-BFGS with different batch sizes.  Though the authors mention prior work arguing that L-BFGS works poorly with stochastic gradient estimates (at least without very large batch sizes), there is no hint as to why stochastic gradients should work so much better with L-SR1.  This seems like a real mystery: is it the L-SR1 Hessian estimate that works better, or the use of ARC rather than line search, or something else entirely?  There are also several aspects of the algorithm that are under-explained in the presence of stochastic estimation.  For example, the step acceptance and regularization adaption are based on an improvement ratio that superficially requires evaluating the full empirical loss at two points; are these replaced by means over a mini-batch in the implementation?  If so, is the same mini-batch used to evaluate both terms?  And similarly with the gradient treatment: is the gradient difference in the L-SR1 algorithm based on stochastic gradient estimates using the same mini-batch, or independent draws?  Also, in the conclusion, the authors mention that they would like to "explore a stochastic version of computing the Hessian approximation", and this confused me in the context of the apparently-stochastic gradients feeding into the L-SR1 Hessian approximation.

Second, the authors never specify the activation function they are using in their experiments.  But if one was to use a ReLU activation (which seems likely), the true empirical risk becomes non-smooth.  There has been interesting work done on using BFGS for non-smooth functions, and it seems to work surprisingly well in that setting; however, to my knowledge, the theory for why is still lacking.  I don't know of any work on SR1 (or L-SR1) for non-smooth objectives.

The paper might still be ready for publication despite missing theoretical details if there were a strong enough empirical justification.  However, while the experiments are interesting, they all involve relatively small networks (only one over 100K parameters) for relatively small data sets, though this might be based on the hardware used (one node with a GPU).

I also have several more minor comments:

- The citation of Luo et al on the first page should be parenthesized.

- The quoted theorem 1 is for convex optimization problem; the start of the next sentence says "Newton's method avoid (sic) these saddle points," but convexity means that saddle points cannot be the problem.

- "Thus storing the Hessian O(n^2) becomes impractical" should be something like "Thus storing the Hessian, which takes O(n^2) memory, becomes impractical."  Similarly "the inversion of the Hessian matrix can make it an additional computational burden O(n^3)" should be something like "the inversion of the Hessian matrix takes O(n^3), which is also impractical" (or just "the inversion of the Hessian matrix takes O(n^3)" -- the reader will likely get the point).

- In Algorithm 3 line 3, the matrix Y_k should have columns y_0 through y_k (vs s_0 through s_k).  Also, I was confused my the update for mu_{k+1} -- it seems like it is being assigned an interval rather than a specific value?

- In the results, please do say more about the architecture (activation functions in particular).  This could go into supplementary details, but the experiments could not be reproduced with what is given here.

- On page 7, "Figure 4(b) tepresents" -> "Figure 4(b) represents"

**Summary Of The Paper:**

The paper describes a limited memory quasi-Newton method based on SR1 updating using a variant on the adaptive regularized cubic (ARC) approach to globalization.  The algorithm is applied to training deep neural networks for image classification and autoencoding, and compared to L-BFGS and various SGD variants.  The authors claim the main contributions to be the different optimizer ingredients (L-SR1, ARC safeguarding, and the particular shape-changing regularizer) as well as computational complexity similar to L-BFGS.

**Summary Of The Review:**

The ingredients of the algorithm are known for smooth, non-noisy functions, though the specific combination seems new.  However, the details of the algorithm are not fully justified even in this setting; and the actual setting of interest seems to involve stochastic function and gradient estimates and possibly non-smooth objectives (depending on the activation).  The experiments that are in the current version are intriguing because they suggest that the algorithm works well (significantly better than L-BFGS) in this setting.  But to accept, I would have liked to see more exploration of these aspects of the problem, either theoretically or empirically.

---

> ### Author Response · Authors · 2021-11-23
> **Official Rebuttal of Paper4406 by Author: Revisions**
>
> We thank the referee for a very thorough review of the paper.
> We have now provided a proof of convergence for the proposed algorithm.  The proof is based on the ARCs paper by Cartis et al (2011) but applied to the SR1 method.  In addition, we have folded Appendix A into the main paper so that the paper could be implemented readily by the reader.
> As the referee pointed out, we have implemented a stochastic version of the proposed algorithm, partly for ease of presentation.  We have now included a short section discussing the stochastic implementation of the proposed algorithm.  We have also modified the conclusion section to reflect this. For the activation functions, we use ReLU. We have included the network architectures in the Appendix section. Although the ReLU function is not differentiable at 0, we did not encounter any difficulties computing the gradient in any of the numerical experiments.

---

### Official Review · Reviewer_qdob · 2021-11-03

**Correctness:** 2
**Technical Novelty And Significance:** 2
**Empirical Novelty And Significance:** 2
**Recommendation:** 3
**Confidence:** 4

**Main Review:**

Although I agree that introducing quasi-newton methods is an interesting direction, this paper seems to be underdeveloped and needs quite a lot of work to be ready for publishing. First of all, there are some errors/missing parts in the paper that seriously need to be corrected.

1. In Algorithm 1, line 4, what is \mathcal{G}? Is it an operator? I believe the authors are trying to use the framework by Reddi et al, 2018, but the notations are not clear enough.

2. The authors claim that the counterexample by Reddi et al. 2018 can be extended to all adaptive optimization algorithms, but that is not true. The counterexample is designed particularly for Adam because its second moment (the denominator) could decrease when optimization progresses, and the telescopic summation in the proof of Adam is thus wrong. I encourage the authors to read the paper of Reddi et al. 2018 more carefully to get a better idea of why Adam is poorly-designed. The other proposed adaptive algorithm, such as AMSGrad, AdaBound (Luo et al, 2019), and NosAdam (Huang et al, 2019) are well-designed and they would not have similar issues.

3. Where is the step size in the definition of newton and quasi-newton methods?

4. I am expecting a theoretical analysis of the convergence behavior of the proposed algorithm, maybe even in the online convex setting to show the advantage of the proposed algorithm. However, I do not see such theorems or propositions.

5. Some of the experimental results are not convincing enough. First of all, I would encourage the authors to run experiments on some larger datasets, such as ImageNet, or on some different tasks apart from image tasks, such as language modeling on PennTreeBank. The current tasks are only image classification/generation, and only on small datasets such as MNIST and CIFAR10, which seem very limited. Moreover, I don't understand why the experiments on CIFAR10 is terminated at the 15-th epoch, while the other algorithms run for as many as 4k epochs. From my experience, 100-150 epochs are pretty sufficient for any algorithm and any NN model to train on CIFAR10. I would like the authors to explain why.

Sashank J. Reddi, Satyen Kale, and Sanjiv Kumar. On the convergence of adam and beyond, ICLR 2018.
Liangchen Luo, Yuanhao Xiong, Yan Liu, and Xu Sun. Adaptive gradient methods with dynamic bound of learning rate, ICLR 2019.
Haiwen Huang, Chang Wang, Bin Dong Nostalgic Adam: Weighting more of the past gradients when designing the adaptive learning rate, IJCAI 2019

**Summary Of The Paper:**

This paper proposes a quasi-newton method for neural network optimization. The authors have conducted some experiments to verify the effectiveness of their algorithm.

**Summary Of The Review:**

I believe this paper still needs a lot of work to be ready for publication.

---

> ### Author Response · Authors · 2021-11-23
> **Official Rebuttal of Paper4406 by Author: Revisions**
>
>    The Authors would like to thank the reviewers for their comments on the paper.
>    1. We have removed the Algorithm 1 as it is trivial. For further understanding, the reader may read Reddi et al (2018) for further clarification.
>    2. We agree with the reviewer and have removed the claims.
>    3. The step size definition is given in the Introduction section right before equation (2) (... that satisfy the secant equation given by y_k = B_{k+1}s_k)
>    4. We agree with the reviewer. We have conducted additional experiments on the ResNet50 network architecture. Due to time-constraints we were not able to perform the experiments on the Imagenet dataset. However, we will include the results from the Imagenet dataset later. The previous iteration of the paper had a few typos and graph errors. We have fixed it in the current iteration of the paper and have uploaded the revised version.

---

### Comment · Reviewer_oGs7 · 2021-11-29
**After rebuttal**

I have read the other reviews and the rebuttals of the authors. The authors agree with the majority of the criticisms including mines. This work is not ready to be published as is. I keep my score unchanged.

---

### Comment · Reviewer_YQSL · 2021-11-29
**After rebuttal**

I thank the authors for their comments.  These partially address my concerns, but the specific concern regarding the convergence proof had to do with the fact that it is a proof in the case of exact gradients (vs the stochastic gradient version).  Having read the revised version and the comments of the other reviewers, I will keep my score where it is.

---

### Comment · Reviewer_qdob · 2021-12-04
**After Rebuttal**

I thank the authors for their rebuttal and clearly, this paper needs a lot more work to be ready for publication. Therefore I will keep my rating. I encourage the authors to resubmit their paper when it is ready.

---

### Decision · Program_Chairs · 2022-01-20

**Decision:**

Reject

**Comment:**

This paper presents an adaptive gradient method for neural net training inspired by L-BFGS. All of the reviewers recommend rejection. They raise concerns about the amount of novelty, the clarity of the writing, and the experimental comparisons. I encourage the authors to take the reviewers' comments into account and improve the submission for the next cycle.